# Mucinous and Non-Mucinous Rectal Adenocarcinoma—Differences in Treatment Response to Preoperative Radiotherapy

**DOI:** 10.3390/jpm10040226

**Published:** 2020-11-13

**Authors:** Karolina Vernmark, Xiao-Feng Sun, Annica Holmqvist

**Affiliations:** 1Department of Biomedical and Clinical Sciences, Linköping University, 58183 Linköping, Sweden; karolina.vernmark@regionostergotland.se (K.V.); Annika.Holmqvist@regionostergotland.se (A.H.); 2Department of Oncology in Linköping, and Department of Surgery, Orthopaedics and Cancer care, Linköping University, 58183 Linköping, Sweden

**Keywords:** mucinous adenocarcinoma, non-mucinous adenocarcinoma, radiotherapy, rectal cancer, survival

## Abstract

There is a need to personalize the treatment for rectal cancer patients. The aim of this study was to analyze therapy response and prognosis after preoperative radiotherapy in rectal cancer patients with mucinous adenocarcinoma compared to those with non-mucinous adenocarcinoma. The study included retrospectively collected data from 433 patients, diagnosed with rectal cancer in the South East health care region in Sweden between 2004 and 2012. Patients with non-mucinous adenocarcinoma that received short-course radiotherapy before surgery had better overall survival, cancer specific survival, and disease-free survival, as well as distant- and local-recurrence-free survival (*p* = 0.003, *p* = 0.001, *p* = 0.002, *p* = 0.002, and *p* = 0.033, respectively) compared to the patients that received long-course radiotherapy with concomitant capecitabine. The results were still significant after adjusting for sex, age, stage, differentiation, and chemotherapy in the neoadjuvant and/or adjuvant setting, except for local-recurrence-free survival that was trending towards significance (*p* = 0.070). In patients with mucinous adenocarcinoma, no difference in survival was seen when comparing patients that had short-course radiotherapy and patients that had long-course radiotherapy. However, none of 18 patients with mucinous adenocarcinoma treated with long-course radiotherapy had local tumor progression, compared to 7% of 67 patients with non-mucinous adenocarcinoma. The results indicate that mucinous adenocarcinoma and non-mucinous adenocarcinoma may respond differently to radiotherapy.

## 1. Introduction

Colorectal cancer (CRC) is the third most common cancer and a major cause of cancer mortality worldwide [1]. A special histology subtype, called mucinous adenocarcinoma (MAC), is defined by the World Health Organization (WHO) as an adenocarcinoma in which at least 50% of the cancer tissue is composed of mucin and accounts for 10–15% of CRCs in large population-based studies [2].

There are several consistent findings concerning differences between colorectal MAC and non-mucinous adenocarcinoma (NMAC). Most notable of these differences are that MAC more often affects female and young patients [3,4], and generally presents at a later stage [3,4,5,6,7]. It has also been shown that MAC, compared to NMAC, is less likely to be resected with negative surgical margins [8,9], more often metastasizes to lymph nodes [10,11], and is more prone to local recurrence [8,9,12] as well as peritoneal carcinomatosis [11,13].

Numerous studies have shown conflicting results regarding the prognosis of colorectal MAC compared to NMAC. Most studies, however, report a worse prognosis for rectal MAC [3,14,15,16] although a large cohort study by Hugen et al. in 2015 [9] showed that from 1999 onwards the survival rates have been comparable, even after adjusting for type of preoperative treatment. Nevertheless, compared to NMAC, MAC has been associated with a poorer response to chemotherapy (CT) and chemo-radiotherapy (CRT) [17,18]. Still, MAC and NMAC patients usually get the same preoperative treatment.

Preoperative radiotherapy (RT) is well known to improve local control for rectal cancer (RC) [19,20]. For stages II-III resectable RCs either preoperative short-course radiotherapy (SCRT) or long-course radiotherapy (LCRT) followed by radical total mesorectal excision (TME) is recommended [21,22].

Although both SCRT and LCRT have been practiced in parallel for more than 20 years, it is not clear which form of preoperative RT provides better tumor control and long-term outcomes. Most studies, however, seem to show that they are equally effective [23,24].

As far as we know, this is the first study that analyzes survival and treatment response in NMAC and MAC patients in relation to preoperative SCRT and LCRT.

## 2. Materials and Methods

### 2.1. Patients

As shown in Figure 1, the study initially included 465 patients from the South East health care region in Sweden that were diagnosed with rectal adenocarcinoma between 2004 and 2013. The study was approved by the Linköping ethical committee (reference number: 2014/39–71), and all patients had given their written consent to participate in studies related to the cancer prior to surgery. A retrospective medical review from each patient’s oncological and surgical files was performed.

### 2.2. Data Collection

Characteristics of the patients and tumors were recorded and included sex, age at diagnosis, tumor location, results from the preoperative magnetic resonance imaging (MRI) of the tumor area and computed tomography of the chest, pathologic tumor stage after surgery, histological type, the grade of differentiation, lymphovascular invasion, as well as type of surgery, preoperative CT, and/or RT. Data also included overall survival (OS), cancer-specific survival (CSS), disease-free survival (DFS), distant-recurrence-free survival (DRFS) and local-recurrence-free survival (LRFS).

### 2.3. Staging and Evaluation of Treatment Effect

Computed tomography of the chest and preoperative MRI were used for staging and, in the case of LCRT, to evaluate the effect of the treatment. The effect was categorized as progressive disease (defined as an increased size of the RC), regression (defined as decreased size of the RC), or stable disease (defined as status quo). Thirteen of the patients had no preoperative MRI or inconclusive MRI and were therefore not included in these analyses.

### 2.4. Treatment

Among 465 patients who had surgery due to RC, we excluded a total of 32 patients, including 27 patients that had no surgery and five patients that had palliative surgery. The 433 remaining patients were included in this study. One hundred and thirty-five patients had no preoperative treatment and nine patients had only preoperative CT without radiotherapy. The 289 (67%) patients that had preoperative radiotherapy included 198 patients (69%) that received SCRT (25.0 Gy in 5.0 Gy fractions) and 91 patients (31%) that received LCRT (44.0–50.4 Gy in 1.8–2.0 Gy fractions). Patients generally had capecitabine concomitant with LCRT, except for seven patients that had only RT. The data are shown in Figure 1. None of the 433 patients had postoperative radiotherapy.

Patients with MAC and NMAC were equally likely to have preoperative RT (72% patients with MAC, 66% patients with NMAC, *p* = 0.361). However, among the patients that had preoperative RT, patients with NMAC more often had SCRT than MAC (71% vs. 51%, *p =* 0.012). The data are shown in Table 1.

Surgeries of the primary RCs were performed with a median of 5 days (range 0–99) after SCRT, and in a median of 44 days after LCRT (range 2–83). Patients that had SCRT generally had surgery within 7 days (85%) while patients that had LCRT generally had surgery after 28–56 days (78%). Most patients had surgery with TME (56%) or abdominoperineal resection (APR) (36%). The remaining patients had either local excision with transendoscopic microsurgery (TEM) (6%) or proctocolectomy (1%). There was no difference in the type of surgery between MAC and NMAC (*p* = 0.718) as shown in Table 1. One hundred and twenty-nine patients (30%) had perioperative CT, either before surgery (neoadjuvant) and/or after surgery (adjuvant), besides the CT given during LCRT. Patients with MAC were more likely to have perioperative CT (*p* < 0.001).

### 2.5. Statistical Analysis

All statistical analyses were conducted with the statistical package Statistica. The Chi-square test was used to examine the relationship of clinicopathological characteristics with treatments. Patients with stage IV disease were excluded in the DFS and DRFS analyses. Survival curves were calculated based on the Kaplan–Meier method. The survival analyses, including univariate and multivariate analyses, were performed with the Cox proportional hazard model. LCRT was used as reference when comparing SCRT to LCRT. The data were presented as hazard ratios (HR) with 95% confidence intervals (CI). Statistical significance was set at *p* < 0.05 for all tests.

## 3. Results

### 3.1. Clinicopathological Characteristics in MAC and NMAC Patients

Mucinous histology was defined as more than 50% mucin content in the postoperative cancer specimen. MAC was found in 54 (12%) of the patients and NMAC in 379 of the patients. MAC patients had a higher TNM stage at the time of diagnosis (predominantly stage III). MAC patients were also less likely to have radical surgery (*p =* 0.005), had a higher T-stage (*p =* 0.008), and more often confirmed lymph node metastases after surgery (*p* = 0.013). There was an overall increased risk for distant recurrence for MAC patients compared to NMAC patients (*p* = 0.009). Besides, there was no significant difference between MAC and NMAC patients concerning sex, age, vascular invasion, perineural growth, or local recurrence (*p* > 0.05). The results are shown in Table 1; Table 2.

### 3.2. Survival in MAC and NMAC Patients

Regarding survival, we first examined patients with MAC compared to patients with NMAC. MAC patients had significantly worse OS (HR 1.59; 95% CI: 1.06–2.38; *p* = 0.026), CSS (HR 2.18; CI 95%: 1.38–3.43; *p* < 0.001), and DRFS (HR 1.93; CI 95%: 1.04–3.60; *p* = 0.038) compared to NMAC patients. There was also a trend towards worse DFS in patients with MAC even though the difference did not reach statistical significance (HR 1.70; CI 95%: 0.94–3.07; *p* = 0.078). However, the significance for OS, CSS, and DRFS was lost in the multivariate analysis after adjusting for sex, age, stage, and differentiation (*p* > 0.05).

We also performed a subgroup analysis for each separate stage (stage I, II, III, and IV, respectively) regarding survival comparison (OS, CSS, DFS, DRFS, LRFS) in MAC patients and NMAC patients. No significant differences were found (*p* > 0.05).

### 3.3. Survival in Relation to Preoperative RT in MAC and NMAC Patients

Next, the relationship between SCRT and LCRT was studied in the MAC and NMAC subgroups, respectively. In the MAC group, no significant differences were found between SCRT and LCRT for OS, CSS (Figure 2a,b), DFS, DRFS, or LRFS (Table 3, *p* > 0.05). In the NMAC group, patients that had received SCRT before surgery had significantly better OS, CSS, DFS, DRFS, and LRFS compared to the group that had received LCRT (HR 0.53; CI 95%: 0.35–0.80; *p* = 0.003, HR 0.43; CI 95%: 0.26–0.72; *p* = 0.001 (Figure 2c,d), HR 0.46; CI 95%: 0.30–0.72; *p* = 0.002, HR 0.45; CI 95%: 0.28–0.72; *p* = 0.002 and HR 0.39; CI 95%: 0.17–0.93, *p* = 0.033, respectively (Table 3). The difference persisted after adjusting for sex, age, stage, differentiation, and perioperative CT (CT in the neoadjuvant or adjuvant setting) in OS, CSS, DFS, and DRFS, and there was a trend towards significance in LRFS (*p* = 0.070) (Table 3).

The survival of the patients that had SCRT or LCRT was also analyzed in the whole group of patients with preoperative RT (*n* = 289). Patients that received SCRT had better OS, CSS, DFS, DRFS, and LRFS (*p* = 0.002, *p* < 0.001, *p* = 0.002, *p* = 0.002 and *p* = 0.028, respectively), compared to patients that had LCRT (Table 3). Even in multivariate analyses, after adjusting for sex, age, stage, differentiation, and perioperative CT, the result in OS, CSS, DFS, and DRFS was still significant and there was a trend towards significance in LRFS (Table 3).

### 3.4. Local Response after LCRT in MAC and NMAC Patients’ Tumors

In 85 patients that had LCRT, the treatment response of the primary RC was evaluated with MRI before and after completion of the LCRT. Comparing the results from the MRI before the start of the treatment with the MRI after completion of the treatment, no difference in number of MRI evaluated treatment responses was seen between MAC and NMAC (*p* = 0.449, Table 1). However, as shown in Table 1, none of 18 MAC patients had progressive disease compared to 7% of 67 of the NMAC patients.

MAC patients that had LCRT and regression of the tumor had better OS (HR 0.21; 95% CI: 0.05–0.89; *p* = 0.033) and CSS (HR 0.18; 95% CI: 0.04–0.79; *p* = 0.023) compared to the patients that did not have regression. This was not seen in the NMAC group, regardless of treatment response, as shown in Figure 3.

## 4. Discussion

According to European Society of Medical Oncology (ESMO) guidelines of 2017, both SCRT and LCRT for locally advanced RC are recommended [21]. The Stockholm III study showed that there was no significant difference in cumulative incidence of local recurrence in patients with delayed surgery after SCRT compared to SCRT with immediate surgery; nor was there a difference in cumulative local recurrence in LCRT with delay compared to any of the SCRT regimens. However, there was an increased rate of postoperative complications in the SCRT group with immediate surgery [25].

To our knowledge, no previous study has analyzed the survival and treatment response to SCRT and LCRT in MAC and NMAC patients specifically. Since we compared survival rather than postoperative complications in the current study, we did not consider the fact that there was a difference between the groups concerning time until surgery.

In this study, we found no difference in treatment response between SCRT and LCRT in MAC patients. However, in NMAC patients, SCRT was related to improved OS, CSS, DFS, DRFS, and LRFS compared to LCRT. The difference in OS, CSS, DFS, and DRFS remained, and for LRFS there was a trend towards significance even when adjusting for sex, age, stage, differentiation grade, and perioperative CT. The value of perioperative (either adjuvant or neoadjuvant) CT for patients with RC is debated since it is often based on extrapolation of results from phase III trials of adjuvant treatment for colon cancer. However, there are several studies that show that both neoadjuvant and adjuvant CT can improve survival for RC as well [26] and therefore it is important to correct for this factor in the multivariate analyses. Since this study showed better survival after SCRT for NMAC patients regardless of tumor stage, it may suggest that SCRT is a better treatment option for NMAC patients than LCRT. These results are important since SCRT has a lower rate of early toxicity compared to LCRT [27,28], and is also less expensive and more convenient both for the patient and for the center [29]. Larger prospective studies are needed to confirm our findings.

There are, however, several factors that indicate that MAC and NMAC are two different types of cancers with different response to RT. Previous studies as well as our study have shown that MAC patients compared to NMAC patients have larger tumors, relatively later-stage presentations, more often have lymph node metastases, and are less likely to have radical surgery [3,14,18,30]. Biologically, it has been shown that MAC tumors are associated with low proliferative activity and less apoptosis [31]. This fact, in combination with the mucus surrounding the tumor cell, which in theory could make it difficult for RT and CT to penetrate into the nucleus and induce apoptosis, could perhaps explain why MAC patients in previous studies have been shown to respond more poorly to CT and CRT compared to NMAC [17,18]. In this study, there was no significant difference between MAC and NMAC patients regarding the number of down-staged tumors after LCRT, but there was another difference in the response to LCRT. When comparing tumor progression during LCRT treatment, we found that none of the MAC tumors progressed compared to 7% of the NMAC tumors. Moreover, while OS, CSS, DFS, and DRFS were better for MAC patients that had LCRT and responded with regression compared to MAC patients that did not, there were no such differences in survival in NMAC patients. The number of patients in this analysis was small; however, this is nevertheless an interesting finding since it may indicate that even though tumor regression was seen after LCRT, LCRT might not be the best treatment for NMAC patients in terms of survival.

Regarding the question of SCRT versus LCRT for RC patients in general, there are two previous meta-analyses that have shown that SCRT and LCRT are equally effective when it comes to OS, DFS, and rates of local recurrence, but that the rate of pathologic complete response is lower with SCRT compared to LCRT [23,24]. The former study by Wang et al. [23] included studies with patients that had locally advanced RC, and the latter by Zhou et al. [24] included studies with RC patients in stages I-III. As far as we know, there is only one previous study that has shown that SCRT is related to better survival compared to LCRT. In this previous Polish phase III trial, 515 patients with cT4 or cT3 tumors were included and further treated with either SCRT followed by consolidation CT or with LCRT with concomitant bolus injections with 5-FU and leucovorin prior to surgery [32]. The study showed that SCRT was related to better three-year OS compared to LCRT. However, the control arm in this study is generally not considered a standard approach for LCRT. The present study included patients of all stages with a potentially curative treatment intention and the patients had LCRT according to ESMO guidelines. Interestingly, the OS, CSS, DFS, and DRFS were better for the whole RC patient group after SCRT compared to LCRT.

For a rectal cancer study, the sample of patients included in this study has the advantage of being a relatively large real-life sample, which represents Swedish conditions well. MAC was present in 12% of the patients, a percentage that is consistent with findings in large population-based studies [2]. The histology used in this study was that of the post-operative specimens. Although it would be interesting to look at the preoperative biopsies to avoid treatment-induced mucinous cancer, mucus commonly coats the tumor at the extraluminal side and tissue in a biopsy is usually taken from the more superficial aspect of the tumor and might not contain a representative amount of mucous [33]. On the other hand, there is of course the risk of including treatment-induced mucinous tumors.

Whether MAC and NMAC patients should be separated before deciding the type of preoperative treatment is still unclear. There is, however, a need to find better treatment options for patients with MAC, considering the relative treatment resistance to CT and RT, and MAC patients would most likely benefit from other types of treatments due to the mucin content and molecular signatures that differ from NMAC.

## 5. Conclusions

The present study demonstrated that patients with NMAC had significantly better survival after SCRT compared to LCRT. No such evidence was seen in MAC patients. The result indicates that the histological subtypes MAC and NMAC have different treatment responses to different preoperative RT. Further larger studies are needed to confirm these findings.

## Figures and Tables

**Figure 1 jpm-10-00226-f001:**
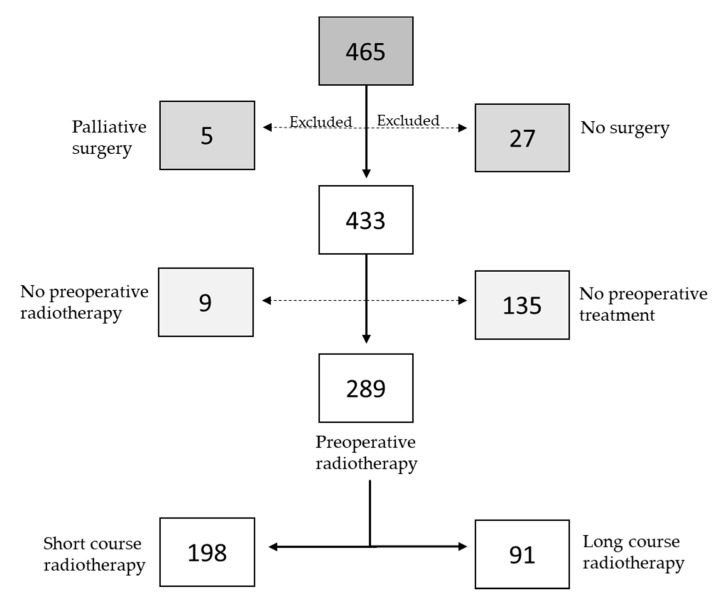
Flow-chart of the patients included in the study and type of preoperative treatment.

**Figure 2 jpm-10-00226-f002:**
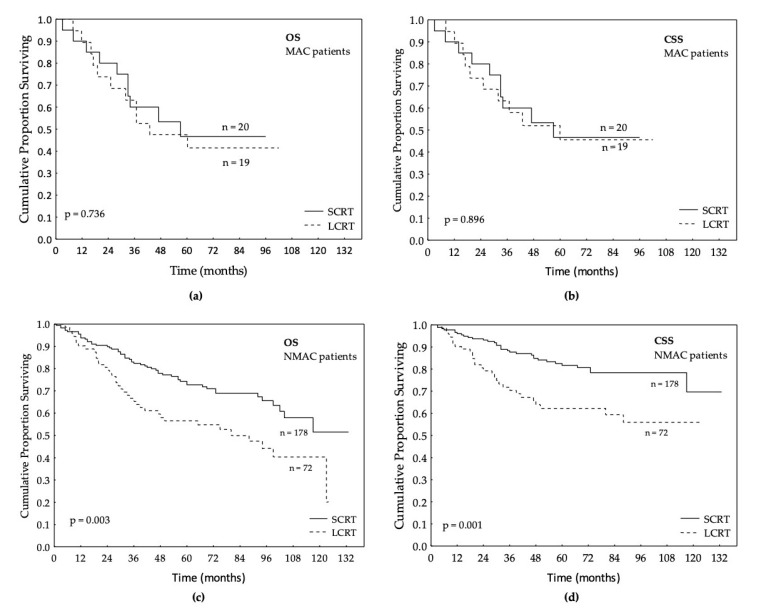
Short-course radiotherapy (SCRT) vs. long-course radiotherapy (LCRT): (**a**) overall survival (OS) in patients with mucinous adenocarcinoma (MAC); (**b**) cancer-specific survival (CSS) in patients with MAC (**c**) overall survival (OS) in patients with non-mucinous adenocarcinoma (NMAC); (**d**) cancer-specific survival (CSS) in patients with NMAC.

**Figure 3 jpm-10-00226-f003:**
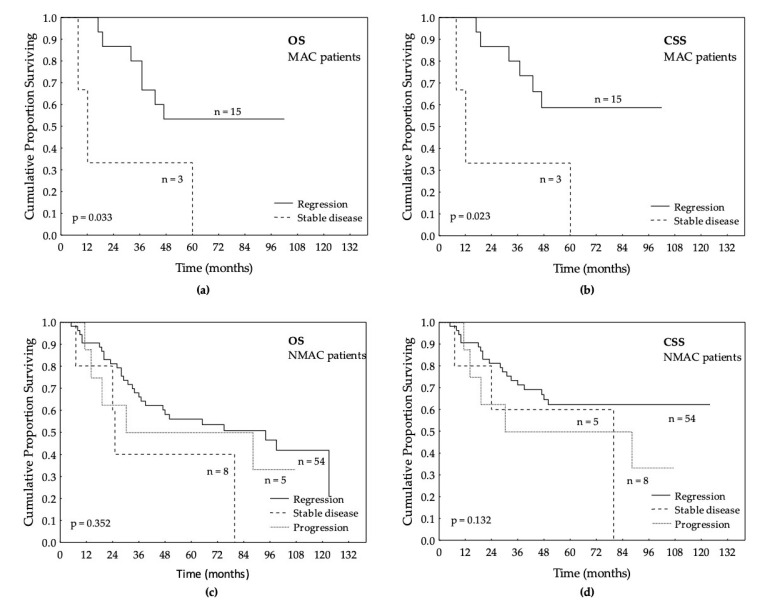
Treatment response after long-course radiotherapy (LCRT): (**a**) overall survival (OS) in patients with mucinous adenocarcinoma (MAC); (**b**) cancer-specific survival (CSS) in patients with MAC; (**c**) overall survival (OS) in patients with non-mucinous adenocarcinoma (NMAC); (**b**) Cancer-specific survival (CSS) in patients with NMAC.

**Table 1 jpm-10-00226-t001:** Different treatments and treatment response in the patients.

	All Patients	MAC	NMAC	*p*-Values ^1^
	*n* = 433 (100%)	*n* = 54 (12%)	*n* = 379 (88%)	
	*n* (%)	*n* (%)	*n* (%)	
Preoperative RT				0.361
Yes	289	39 (72)	250 (66)	
No	144	15 (28)	129 (34)	
Type of RT				0.012
SCRT	198	20 (51)	178 (71)	
LCRT	91	19 (49)	72 (29)	
Response after LCRT	85	18	67	0.449
Regression (partial or complete)	69	15 (83)	54 (81)	
Stable disease	11	3 (17)	8 (12)	
Progression	5	0	5 (7)	
Type of surgery				0.718
APR	158	22 (41)	136 (36)	
TEM	28	2 (4)	26 (7)	
TME	242	29 (54)	213 (56)	
Proctocolectomy	5	1 (2)	4 (1)	
Resection margin				0.005
No tumor cells (R0)	396	44 (81)	352 (93)	
Remaining tumor cells (R1)	37	10 (19)	27 (7)	
Distance to anal verge (cm)				0.815
Mean (cm)	8.0	7.5	8.1	
Perioperative CT				<0.001
Yes	129	27 (50)	102 (27)	
No	304	27 (50)	277 (73)	
Local recurrence				0.796
Yes	36	4 (7)	32 (8)	
No	397	50 (93)	347 (92)	
Distant recurrence				0.009
Yes	120	23 (43)	97 (26)	
No	313	31 (57)	282 (74)	

APR = abdominal perineal resection, MAC = mucinous adenocarcinoma, NMAC = non-mucinous adenocarcinoma, RT = radiotherapy, SCRT = short-course radiotherapy, LCRT = long course radiotherapy, TEM = transendoscopic microsurgery, TME = total mesorectal excision, Perioperative CT = perioperative chemotherapy, i.e., chemotherapy in the neoadjuvant (preoperative) and/or adjuvant (postoperative) setting. ^1^
*p*-values calculated by Chi-square test compare MAC patients and NMAC patients.

**Table 2 jpm-10-00226-t002:** Clinicopathological characteristics in the patients.

	All Patients	MAC	NMAC	*p*-Values ^1^
	*n* = 433 (%)	*n* = 54 (%)	*n* = 379 (%)	
Sex				0.785
Male	266	30 (56)	248 (58)	
Female	185	24 (44)	161 (42)	
Age (years)				0.267
≤69	202	29 (54)	173 (46)	
>69	231	25 (46)	206 (54)	
TNM stage				0.015
I	129	7 (13)	122 (32)	
IIA	110	10 (19)	100 (26)	
IIB	10	2 (4)	8 (2)	
IIIA	17	2 (4)	15 (4)	
IIIB	57	11 (20)	46 (12)	
IIIC	56	11 (20)	45 (12)	
IV	54	11 (20)	43 (11)	
T-stage				0.008
T1	49	0	49 (13)	
T2	113	11 (20)	102 (27)	
T3	230	35 (65)	195 (52)	
T4	38	8 (15)	30 (8)	
Unknown	3	0	3 (1)	
N-stage				0.013
N0	244	21 (41)	223 (63)	
N1	86	15 (29)	71 (20)	
N2	77	15 (29)	62 (17)	
Unknown	26	3 (6)	23 (6)	
Vascular invasion				0.676
Yes	87	12 (22)	75 (20)	
No	346	42 (78)	304 (80)	
Perineural growth				0.340
Yes	58	5 (9)	53 (14)	
No	375	49 (91)	326 (86)	
Differentiation				<0.001
Well	42	1 (2)	41 (11)	
Moderately	287	19 (35)	251 (66)	
Poorly	132	34 (63)	87 (23)	

MAC = mucinous adenocarcinoma, NMAC = non-mucinous adenocarcinoma. ^1^
*p*-values calculated by Chi-square test and compare MAC patients and NMAC patients.

**Table 3 jpm-10-00226-t003:** Patient survival after short-course radiotherapy vs. long-course radiotherapy.

		All Patients		
	Univariate		Multivariate	
	HR ^1^ (95% CI)	*p*-value	HR ^1^ (95% CI)	*p*-value
Overall survival	0.55 (0.38–0.80)	0.002	0.59 (0.41–0.87)	0.007
Cancer-specific survival	0.47 (0.30–0.73)	<0.001	0.57 (0.37–0.90)	0.016
Disease-free survival	0.54 (0.36–0.80)	0.002	0.62 (0.41–0.93)	0.021
Distant-recurrence-free survival	0.52 (0.34–0.79)	0.002	0.58 (0.38–0.90)	0.014
Local-recurrence-free survival	0.41 (0.18–0.91)	0.028	0.48 (0.21–1.08)	0.076
		NMAC		
	Univariate		Multivariate	
	HR ^1^ (95% CI)	*p*-value	HR ^1^ (95% CI)	*p*-value
Overall survival	0.53 (0.35–0.80)	0.003	0.57 (0.37–0.88)	0.011
Cancer-specific survival	0.43 (0.26–0.72)	0.001	0.52 (0.30–0.88)	0.014
Disease-free survival	0.46 (0.30–0.72)	<0.001	0.46 (0.29–0.74)	0.001
Distant-recurrence-free survival	0.45 (0.28–0.72)	<0.001	0.44 (0.27–0.72)	0.001
Local-recurrence-free survival	0.39 (0.17–0.93)	0.033	0.45 (0.19–1.07)	0.070
		MAC		
	Univariate		Multivariate	
	HR ^1^ (95% CI)	*p*-value	HR ^1^ (95% CI)	*p*-value
Overall survival	0.86 (0.37–2.03)	0.736	0.50 (0.19–1.30)	0.157
Cancer-specific survival	0.94 (0.39–2.27)	0.896	0.54 (0.28–2.34)	0.209
Disease-free survival	1.55 (0.62–3.86)	0.347	1.22 (0.45–3.29)	0.696
Distant-recurrence-free survival	1.49 (0.60–3.72)	0.393	1.33 (0.50–3.52)	0.572
Local-recurrence-free survival	0.49 (0.04–5.36)	0.555	- ^2^	- ^2^

^1^ Long-course radiotherapy is reference. Covariates entered in the multivariate model were sex, age, TNM stage, the grade of differentiation and perioperative chemotherapy. ^2^ Multivariate analysis was unavailable due to too few cases.

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
