# Peer review of "Mucinous and Non-Mucinous Rectal Adenocarcinoma—Differences in Treatment Response to Preoperative Radiotherapy"

_jpm, 2020, doi:10.3390/jpm10040226_

Round 1

Reviewer 1 Report

The authors present an interesting retrospective study on the utility of short and long-term radiation therapy on mucinous and non-mucinous rectal adenocarcinomas. The writing was very clear and presents the data in a way that the points are easily understandable. I only had a few minor points for the authors to consider which are listed below.

Materials and Methods

2.1 Patients

If there is additional specific information regarding which governing body and/or institutional review board reviewed the study, it would be good to include that.

Line 94

"Flow-chart presents" should be "Flow-chart of".

Author Response

If there is additional specific information regarding which governing body and/or institutional review board reviewed the study, it would be good to include that.

Response: Thank you very much for your valuable feedback. We have carefully revised the manuscript following the comments and resubmitted a revised manuscript (changes are both tracked and highlighted in grey). 2.1 Patients: We have added the name of the review board that reviewed the study (Linköping ethical committee) and the number of the application on Page 2 Line 66.

 Line 94 "Flow-chart presents" should be "Flow-chart of".

Response: Line 94 (now Page 3 Line 102). We have changed "Flow-chart presents" to "Flow-chart of".

Reviewer 2 Report

This manuscript is an original article that retrospectively evaluated therapy response and prognosis after preoperative radiotherapy in rectal cancer patients with mucinous adenocarcinoma compared to non-mucinous adenocarcinoma. The authors demonstrated preoperative short-course radiotherapy had better benefit for survival in non-mucinous adenocarcinoma compared to preoperative long-course radiotherapy with concomitant capecitabine. Furthermore, different response to radiotherapy between mucinous and non-mucinous adenocarcinoma was indicated based on the result in local progression rate after long-course radiotherapy.

This study was conducted well, and the methods are appropriate. The data are presented clearly. In general, this is a well-written paper that presents interesting data. The results will be of interest to clinicians in the field.

However, the following major and minor issues require clarification:

Major

  1. Postoperative chemotherapy or additional radiotherapy can influence the clinical course and prognosis. The authors should describe and discuss these treatments after surgery.

Minor

  1. (Table 2) The authors should replace “Good” with “Well”.
  2. (P6L157-158) P-values did not correspond with those described in Figure 2c-d.

Author Response

Major

  1. Postoperative chemotherapy or additional radiotherapy can influence the clinical course and prognosis. The authors should describe and discuss these treatments after surgery.

 Response: Thank you very much for your valuable feedback. We have carefully revised the manuscript  following the comments and resubmitted a revised manuscript (changes highlighted in grey).

  • Following your great suggestions, we have added chemotherapy in the adjuvant (postoperative) or neoadjuvant (preoperative) setting as a factor to run new multivariate analyses. As we made all analyses again, we discovered that we had made a typing error in the HR and p-value in the previous manuscript for the univariate analyses of DFS and DRFS in All patients, NMAC and MAC. The new and correct numbers are now shown in Table 3. The corresponding changes have also been made in the Results, Page 6 Line 169-175, Page 8 Line 231-232 and Line 234 as well as in the Discussion in Page 9 Line 368-374 and in Page 10 Line 377, Line 394 and Line 411. We have also added information about the perioperative chemotherapy in Page 4 Line 121-123 and in Table 1 and it’s corresponding footnote in Page 4 Line112-113 as well as the footnote for Table 3 in Page 8 Line 227. The Abstract is now updated (Page 1 Line 17-22).
  • We added a new reference (number 26) to the Reference list, in relation to the added discussion in Page 9-10 Line 370-377.
  • None of the patients had postoperative radiotherapy. This has now been stated in Page 2 Line 97.

Minor

  1. (Table 2) The authors should replace “Good” with “Well”.
  2. (P6L157-158) P-values did not correspond with those described in Figure 2c-d.

Response:

  1. We have changed “Good” to “Well” , and accordingly changed “moderate” to “moderately”, “poor” to “poorly” in Table 2.
  2. P-values for OS and CSS in Page 6 Line 170-171 (previously P6L157-158) have been changed to the correct P-values presented in Table 3 and Figure 2.

 Other adjustments made in the manuscript:

  • The word “chemotherapy” has been changed to “CT” in Page 2 Line 93 and in Page 10 Line 340.

In Page 4 Line 120: p = .718 has been changed to p = 0.718

Round 2

Reviewer 2 Report

The manuscript is much improved enough to be accepted.